# Diagnosing the Climatic and Agronomic Dimensions of Rain-Fed Oat Yield Gaps and Their Restrictions in North and Northeast China

**Chong Wang [1,2], Jiangang Liu [3], Shuo Li [1,2], Ting Zhang [1], Xiaoyu Shi [1,2], Zhaohai Zeng [1], Yongdeng Lei [1,2,*] and Qingquan Chu [1,2,*]**

[1] College of Agronomy and Biotechnology, China Agricultural University, Beijing 100193, China; chongw@cau.edu.cn (C.W.); lis@cau.edu.cn (S.L.); zhangting0426@163.com (T.Z.); shixiaoyu45@163.com (X.S.); zengzhaohai@cau.edu.cn (Z.Z.)

[2] Key Laboratory of Farming System, Ministry of Agriculture and Rural Affairs, Beijing 100193, China

[3] The Institute of Vegetables and Flowers Chinese Academy of Agricultural Sciences, Beijing 100081, China; liujiangang@caas.cn

[*] Correspondence: leiyd@cau.edu.cn (Y.L.); cauchu@cau.edu.cn (Q.C.)

**Abstract:** Confronted with the great challenges of globally growing populations and food shortages, society must achieve future food security by increasing grain output and narrowing the gap between potential yields and farmers' actual yields. This study attempts to diagnose the climatic and agronomic dimensions of oat yield gaps and further to explore their restrictions. A conceptual framework was put forward to analyze the different dimensions of yield gaps and their limiting factors. We quantified the potential yield (*Yp*), attainable yield (*Yt*), experimental yield (*Ye*), and farmers' actual yield (*Ya*) of oat, and evaluated three levels of yield gaps in a rain-fed cropping system in North and Northeast China (NC and NEC, respectively). The results showed that there were great differences in the spatial distributions of the four kinds of yields and three yield gaps. The average yield gap between *Yt* and *Ye* (YG-II) was greater than the yield gap between *Yp* and *Yt* (YG-I). The yield gap between *Ye* and *Ya* (YG-III) was the largest among the three yield gaps at most sites, which indicated that farmers have great potential to increase their crop yields. Due to non-controllable climatic conditions (e.g., light and temperature) for obtaining *Yp*, reducing YG-I is extremely difficult. Although YG-II could be narrowed through enriching soil nutrients, it is not easy to improve soil quality in the short term. In contrast, narrowing YG-III is the most feasible for farmers by means of introducing high-yield crop varieties and optimizing agronomic managements (e.g., properly adjusting sowing dates and planting density). This study figured out various dimensions of yield gaps and investigated their limiting factors, which should be helpful to increase farmers' yields and regional crop production, as long as these restrictions are well addressed.

**Keywords:** rain-fed oat; potential yield; yield gaps; limiting factors; agronomic management; food security

## 1. Introduction

The world population is projected to reach 9 billion by 2050 [1], which will become a significant problem for global food security. Subject to the declining cultivated area, it is unpractical to expand farmland for increasing crop yields [2]. However, it is feasible to increase crop production by narrowing the yield gaps between different producers in different regions. Actually, it is possible to increase yields by 45% to 70% for most crops by means of improving nutrient management and increasing irrigation [3], and it is extremely urgent that these yield gaps are narrowed. Oat, a cereal crop with very

high nutritional value, is mainly distributed in Russia, Canada, China, the European Union, Australia, and the USA [4]. Particularly in North China (NC) and Northeast China (NEC), oat is an important crop for local farmers, accounting for over 80% of the country's total planting area of oat. However, oat is mainly grown in arid or semi-arid areas with relatively poor soil and irrigation conditions in China. The unfavorable climatic conditions, poor soil fertility, as well as the inadequate agronomic management have resulted in a low level of farmers' actual yields.

Yield gaps are generally defined as the differences between various levels of crop yields [5]. There are many kinds of yield levels, such as potential yield, attainable yield, water-and-nitrogen limited yield, on-farm yield, and so forth. In general, potential yield can be defined as the yield of an adapted crop variety when grown without water, nutrients, pests, or diseases limiting [6,7]. Potential yield is the yield ceiling of crops [8], which is mainly determined by solar radiation and temperature in irrigated systems [9,10], while in rain-fed systems, the potential yield is mainly limited by solar radiation, temperature, and precipitation. In addition, conventional breeding can realize the continuous progress of potential yield, which is of great significance to yield improvement at the farm level. A study has shown that increased genetic potential yield of maize and wheat varieties is related to a gradual expansion of the genetic background, and there is little evidence of such slowdown [8]. In recent decades, a variety of studies focused on the quantification of potential yield, and the specific methods include field experiments, yield contests, and model simulations [11]. Crop model has become an important tool to simulate crop potential yield. For example, based on the MONICA model simulation in southern Amazon, the combination of high temperature and low rainfall results in the difference between potential yield and water-limited yield of maize and cotton [12]. Crop models not only have obvious advantages of studying potential yield in different agroecological areas but can also simulate the effects of climatic factors and agronomic practices on yield gap [13].

More researchers have concentrated on exploring the limiting factors of yield gaps during the past several years, in order to investigate possible ways to narrow yield gaps [14,15]. Although there are many factors affecting yield gaps, the approaches to reduce yield gaps mainly include the reasonable improvement of agronomic practices and making full use of climatic conditions [16,17]. In addition, compared with some major crops such as rice, maize, and wheat, research on the yield gaps of oat has rarely been reported. In this study, we examined oat yield gaps under a rain-fed system by combining the methodologies of crop models, field experiments, and statistical analysis. The objectives of this study were: (a) To analyze the spatial variations in potential yield, attainable yield, experimental yield, and actual yield of rain-fed oat in NC and NEC, (b) to quantify different levels of yield gaps in NC and NEC, and (c) to diagnose the limiting factors of oat yield gaps, so as to search for effective approaches to narrow yield gaps.

## 2. Materials and Methods

### 2.1. Study Area

The NC and NEC are the typical planting areas of rain-fed oat. The study areas are located in arid and semi-arid climatic zones and their soil types are mainly brown and black soil. Oat in these regions is cultivated in the farmland with relatively poor soil fertility. Local cold weather conditions are suitable for oat growth. In consideration of the diversified agro-ecological conditions, we selected 10 sites from National Oat Industry Technology System in China for our case studies, they are: Youyu (39.98° N, 112.47° E), Zhangjiakou (40.82° N, 114.88° E), Hohhot (40.83° N, 111.73° E), Chahar (40.78° N, 113.22° E), and Datong (40.08° N, 113.30° E) Cities in NC and Baicheng (45.62° N, 122.83°E ), Harbin (45.80° N, 126.53° E), Nenjiang (49.17° N, 125.23° E), Daqing (46.58° N, 125.03° E), and Ulanhot (46.08° N, 122.05° E) Cities in NEC (Figure 1). The climatic conditions during rain-fed oat growing season at these 10 sites in NC and NEC were presented. Average temperature, maximum temperature, minimum temperature, precipitation, solar radiation, and sunshine duration in NC ranged from 14.36 to 17.69 °C, from 21.17 to 24.55 °C, from 6.19 to 10.42 °C, from 94.25 to 141.23 mm, from 1794.41

to 2105.15 MJ m$^{-2}$, and from 727.53 to 876.07 h, respectively. Similarly, average temperature, maximum temperature, minimum temperature, precipitation, solar radiation, and sunshine duration in NEC varied between 16.46 and 17.99 °C, between 22.45 and 24.36 °C, between 10.14 and 11.29 °C, between 172.75 and 286.74 mm, between 1861.00 and 2277.21 MJ m$^{-2}$, and between 760.87 and 942.76 h, respectively.

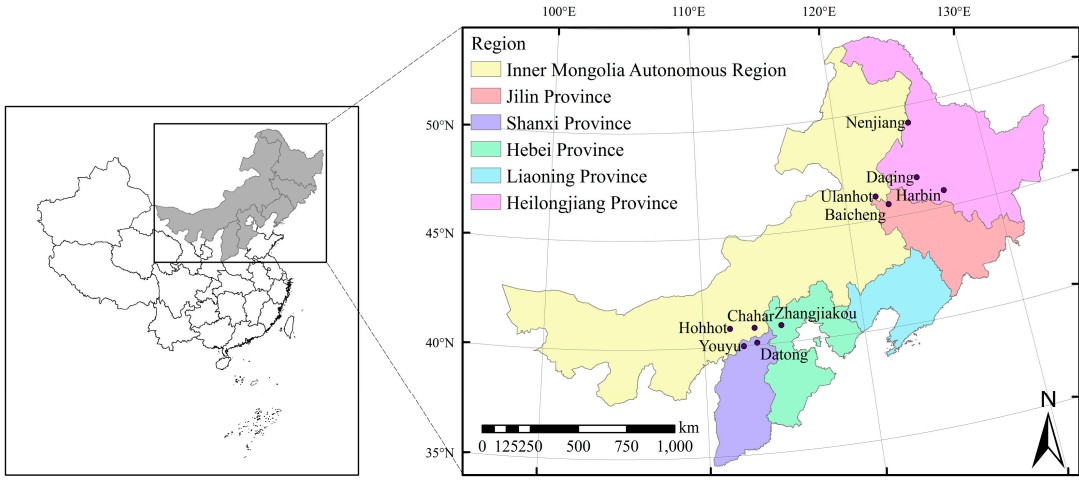

**Figure 1.** The 10 sites selected for case studies of rain-fed oat in North and Northeast China.

### 2.2. Method of Evaluating Potential Productivity

We used the well-known AEZ (Agro-Ecological Zones) model proposed by FAO (Food and Agriculture Organization of the United Nations) [18,19] to calculate light-temperature potential productivity of oat based on climate data in NC and NEC.

In this study, the light–temperature potential productivity is defined as the maximum yield determined only by light and temperature factors without limitation of fertilizer in rain-fed cropping systems. This light–temperature potential productivity of oat can be calculated by Equation (1):

$$
\begin{cases}
Y = CL \times CN \times CH \times G[F(0.8 + 0.011ym)yo + (1 - F)(0.5 + 0.025ym)yc], \ ym > 20 \ \text{kg}/\left(\text{ha}^2{\cdot}\text{h}\right) \\
Y = CL \times CN \times CH \times G[F(0.5 + 0.025ym)yo + (1 - F)0.05ymyo], \ ym < 20 \ \text{kg}/\left(\text{ha}^2{\cdot}\text{h}\right)
\end{cases} \tag{1}
$$

where $Y$ is light-temperature potential productivity; $CL$ is the correction coefficient of leaf area index; $CN$ is the correction coefficient of net dry matter production; $CH$ is the harvest index; $G$ is the whole growth period; $F$ is the cloud coverage ratio; $ym$ is the dry matter production ratio; $yo$ is total dry matter production on cloudy days; and $yc$ is total dry matter production on sunny days.

Meteorological potential productivity is defined as the maximum potential yield that crops can achieve under favorable light, temperature, and limited precipitation conditions. It can be calculated by Equation (2):

$$
Yp = Y \times I_y \tag{2}
$$

where $Yp$ is meteorological potential productivity and $I_y$ is the correction coefficient of moisture. $Iy$ can be calculated using Equations (3) and (4):

$$
I_y = \begin{cases} 1 - K_y \times \left(1 - \frac{ET_a}{ET_m}\right) & , \ ET_a < ET_m \\ 1, \ ET_a > ET_m \end{cases} \tag{3}
$$

$$
ET_m = K_c \times ET_o \tag{4}
$$

where $K_y$ is the yield reflection coefficient; $ET_a$ is actual crop evapotranspiration; $ET_m$ is the water requirement; $K_c$ is the crop coefficient; and $ET_o$ is reference crop evapotranspiration.

## 2.3. Definitions and Quantification of Yield Gaps

We put forward a conceptual framework to analyze the different dimensions of yield gaps, in which three kinds of yield gaps were included (Figure 2). These three yield gaps were composed of four different yield levels, which were potential yield, attainable yield, experimental yield, and farmers' actual yield.

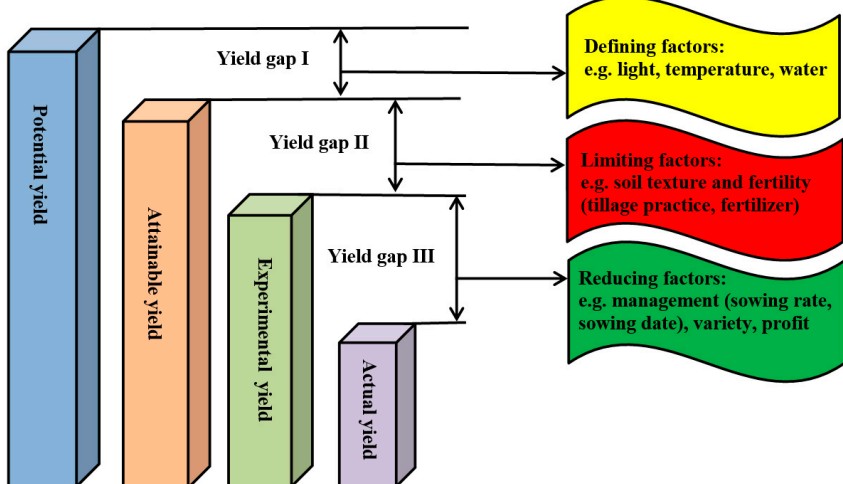

**Figure 2.** A conceptual framework for indicating different yield gaps and their limiting factors. Yield gap I is the difference between potential yield and attainable yield. Yield gap II is the difference between attainable yield and experimental yield. Yield gap III is the difference between experimental yield and farmers' actual yield.

Given the rain-fed environment of oat growth, the potential yield ($Yp$) was the meteorological potential productivity calculated based on the AEZ model. Potential yield obtained under optimal conditions of solar radiation and temperature is only affected by water, and potential yield is the maximum yield under non-irrigated systems in study sites. Attainable yield ($Yt$), the estimated meaningful $Yp$ level of exploitation ceiling by farmers, was the yield obtained by high-yield practices [20]. The definition of $Yt$ was slightly different for different scientists and discussed by many scholars. Lobell et al. [11] used 80% of potential yield as the attainable yield, while Xu et al. [21] defined attainable yield as the maximum yield under local climatic conditions. In this study, we adopted the highest experimental yield as attainable yield based on better local technology and crop management practices in a rain-fed cropping system. The experimental yield was derived from experiments at the 10 study sites from 2011 to 2015. We selected the maximum experimental yield in five years as the attainable yield of each experiment station.

Experimental yield ($Ye$) can be considered as the annual average yield in field experiments, which indicates mean productivity levels of crops in an experiment under local climatic conditions, better management measures, and popularized high-yielding varieties. Agronomic practices at the experiment stations include more reasonable fertilizer use, more effective pest and weed control, and better planting density compared with local management practices. Besides, the varieties used at the experiment stations are the higher yielding varieties in contrast to varieties grown by farmers. Actual yield ($Ya$) is the annual average yield from the statistical data collected by local agricultural investigators. This indicator reflects the farmers' yield obtained under local climatic conditions, soil, varieties, and farmers' actual management practices. Extensive management is one of the most important features of agronomic measures for local farmers. Moreover, there are great differences in varieties that are used in different fields.

Based on the above definitions, we quantified three kinds of yield gaps as follows:

$$\text{YG-I} = Yp - Yt$$
$$\text{YG-I percentage} = [(Yp - Yt)/Yp] \times 100\% \tag{5}$$

$$\text{YG-II} = Yt - Ye$$
$$\text{YG-II percentage} = [(Yt - Ye)/Yt] \times 100\% \tag{6}$$

$$\text{YG-III} = Ye - Ya$$
$$\text{YG-III percentage} = [(Ye - Ya)/Ye] \times 100\% \tag{7}$$

Yield gap I (YG-I) was estimated as the difference between *Yp* and *Yt*. Yield gap II (YG-II) was evaluated as the difference between *Yt* and *Ye*. Yield gap III (YG-III) was calculated as the difference between *Ye* and *Ya* (Figure 2). The YG-I percentage (a value from 0% to 100%) indicates how close *Yt* is to *Yp*. YG-II percentage (a value from 0% to 100%) indicates how close *Ye* is to *Yt*. YG-III percentage (a value from 0% to 100%) indicates how close *Ya* is to *Ye*.

### 2.4. Data Collection

The meteorological data required by the AEZ model are from the National Meteorological Information Centre of China Meteorological Administration (CMA) from 2011 to 2015. These data included average daily temperature, maximum temperature, minimum temperature, mean wind speed, precipitation, solar radiation, sunshine duration, etc. The crop data are obtained from the modern farming system networks in China and field experiments in the study area, which included sowing date, harvest date, leaf area index, and harvest index of oat.

The attainable yields and experimental yields are from 10 experiment stations of National Oat Industry Technology System in NC and NEC. During the years 2011–2015, China launched a high-yield trial of oat in different ecological zones, which was designed with identical crop varieties and management practices on each selected experiment station. The 90 kg/ha of nitrogen fertilizer was applied, of which 60% was applied as a base fertilizer and the other 40% was applied during the trefoil period. The 45 kg/ha of phosphorus fertilizer and 45 kg/ha of potassium fertilizer were applied as base fertilizers. Row spacing was 30 cm, depth of seeding was 5 cm, and sowing rate was 150 kg/ha. The chemical pesticide was mixed into seed to prevent pests and diseases. After oats were sown, the herbicide is sprayed on the fields. There was no irrigation during the whole growth period, and the water demand of oats relied entirely on precipitation. Soil properties, sowing date, and harvest date in selected locations are summarized in Table 1.

**Table 1.** Soil properties, sowing date, and harvest date of oat in different study sites.

| Sites | Soil Types | Total-N (g/kg) | Available P (mg/kg) | Available K (mg/kg) | pH | Sowing Date | Harvest Date |
|-------|-----------|----------------|---------------------|---------------------|-----|-------------|--------------|
| Youyu | Chestnut soil | 0.50 | 27.41 | 88.00 | 8.23 | 15 April | 15 July |
| ZhangJiakou | Chestnut soil | 1.06 | 13.10 | 118.00 | 7.90 | 5 April | 25 June |
| Hohhot | Chestnut soil | 0.53 | 6.68 | 105.20 | 8.47 | 15 April | 13 July |
| Chahar | Chestnut soil | 0.52 | 10.50 | 90.80 | 8.30 | 15 April | 13 July |
| Datong | Chestnut soil | 0.89 | 5.66 | 71.00 | 8.43 | 15 April | 13 July |
| Baicheng | Chernozem soil | 0.80 | 9.90 | 76.60 | 8.20 | 10 April | 15 July |
| Harbin | Black soil | 0.98 | 18.32 | 109.27 | 7.50 | 10 April | 15 July |
| Nenjiang | Black soil | 1.14 | 28.37 | 154.50 | 8.24 | 25 April | 10 August |
| Daqing | Black soil | 0.74 | 11.40 | 94.70 | 8.30 | 10 April | 15 July |
| UlanHot | Aeolian sandy soil | 0.37 | 3.81 | 78.51 | 8.00 | 22 April | 15 July |

The actual yield is from the National Bureau of Statistics of China, which reflects the average productivity in farmers' fields in NC and NEC from 2011 to 2015. Agronomy practices in farmers' actual production are usually extensive, and there are great differences in management measures among different farmers. All fertilizers are applied before sowing with nitrogen fertilizer 120–160 kg/ha, phosphorus fertilizer 60–90 kg/ha, potassium fertilizer 60–80 kg/ha, and no extra fertilizer is applied

during the oat growth period. The sowing date is similar to that in the experimental stations. The sowing rate typically ranges from 200 to 300 kg/ha according to local traditions. Most farmers do not take weeding measures for crop protection. There is no irrigation in farmers' fields. Other management procedures follow local agricultural practices.

## 3. Results

### 3.1. Potential Yield

The regional mean *Yp* of oat was 4304 kg/ha and 4389 kg/ha in NC and NEC, respectively. *Yp* ranged from 3680 kg/ha to 5320 kg/ha in NC, and from 3830 kg/ha to 4863 kg/ha in NEC among the different sites (Table 2). The average value of *Yp* in NEC was slightly higher than that in NC owing to more precipitation and solar radiation in NEC. Spatially, the value of *Yp* decreased from south to north and from east to west, and higher *Yp* values were distributed in NEC and lower values were located in NC (Figure 3). The higher values of *Yp* in NEC were associated with more favorable factors (e.g., stronger solar radiation, longer sunshine duration, and more precipitation during the growth seasons of oat), despite temperature in NEC was lower than that in NC. As a result, higher *Yp* values were mostly found in the sites of NEC (Harbin, Nenjiang, and Daqing Cities with more than 4389 kg/ha) and lower *Yp* values were mainly distributed in the stations of Youyu, Datong, and Chahar Cities of NC with less than 4304 kg/ha (Figure 4). The spatial variation in *Yp* between NC and NEC was the result of the combined effects of stronger solar radiation, longer sunshine duration in NEC with the uneven precipitation, and higher temperature in NC.

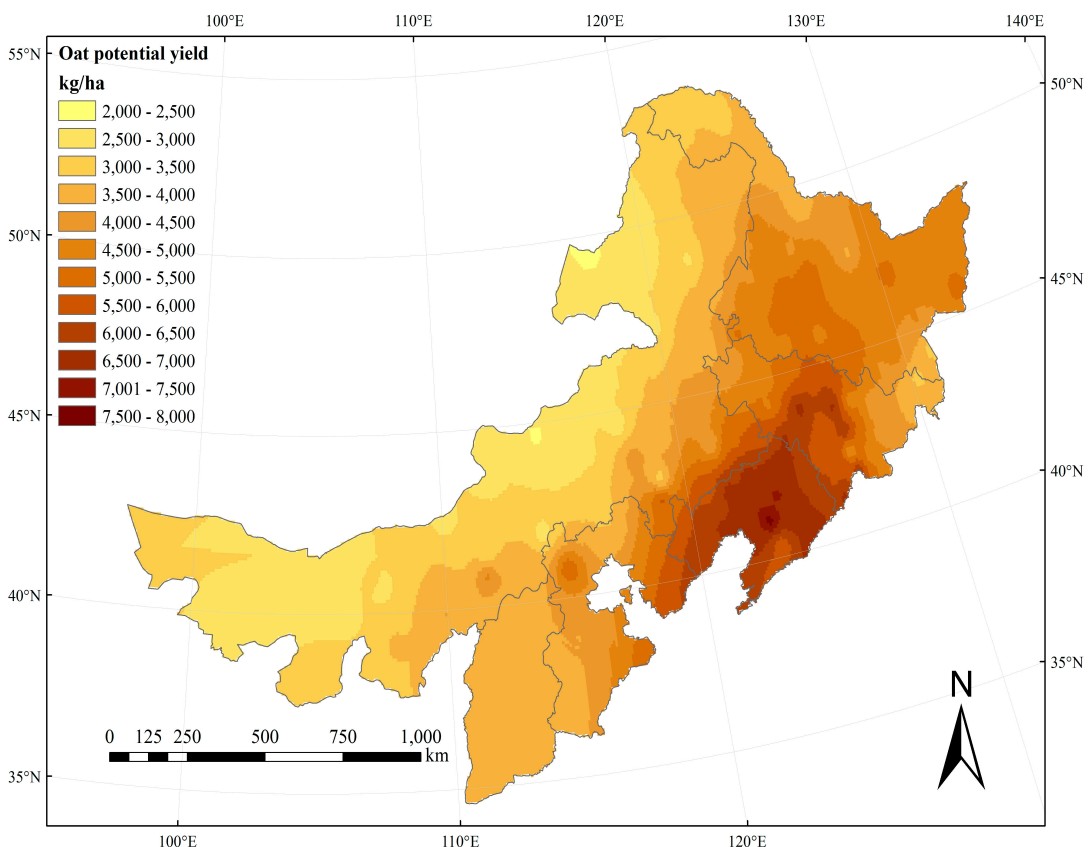

**Figure 3.** The spatial pattern of potential yield for rain-fed oat in North and Northeast China.

**Table 2.** Mean values, maximum values, and minimum values of yield, yield gap, and yield gap percentage for rain-fed oats in North and Northeast China. (*Yp*, potential yield; *Yt*, attainable yield; *Ye*, experimental yield; *Ya*, actual yield; YG-I, yield gap I; YG-II, yield gap II; YG-III, yield gap III; YG-I%, yield gap I percentage; YG-II%, yield gap II percentage; YG-III %, yield gap III percentage; NC, North China; NEC, Northeast China.).

|  |  | Mean | | Maximum | | Minimum | |
|---|---|---|---|---|---|---|---|
|  |  | NC | NEC | NC | NEC | NC | NEC |
| Yield (kg/ha) | *Yp* | 4304 | 4389 | 5320 | 4863 | 3680 | 3830 |
|  | *Yt* | 3369 | 3571 | 3691 | 3786 | 2916 | 3416 |
|  | *Ye* | 2317 | 2396 | 2490 | 2602 | 2034 | 2265 |
|  | *Ya* | 989 | 1656 | 1230 | 1800 | 795 | 1462 |
| Yield gap (kg/ha) | YG-I | 935 | 818 | 1697 | 1182 | 643 | 285 |
|  | YG-II | 1051 | 1175 | 1303 | 1436 | 786 | 814 |
|  | YG-III | 1328 | 740 | 1443 | 944 | 1197 | 551 |
| Yield gap percentage | YG-I% | 21% | 18% | 32% | 25% | 16% | 7% |
|  | YG-II% | 31% | 33% | 39% | 38% | 24% | 24% |
|  | YG-III% | 58% | 31% | 62% | 39% | 50% | 23% |

Note: The mean value for each type of yield is calculated as arithmetic mean of yield among the study sites with similar geographical conditions from NC and NEC, respectively. The maximum and minimum for each type of yield refer to the highest and lowest yields among the study sites from NC and NEC, respectively.

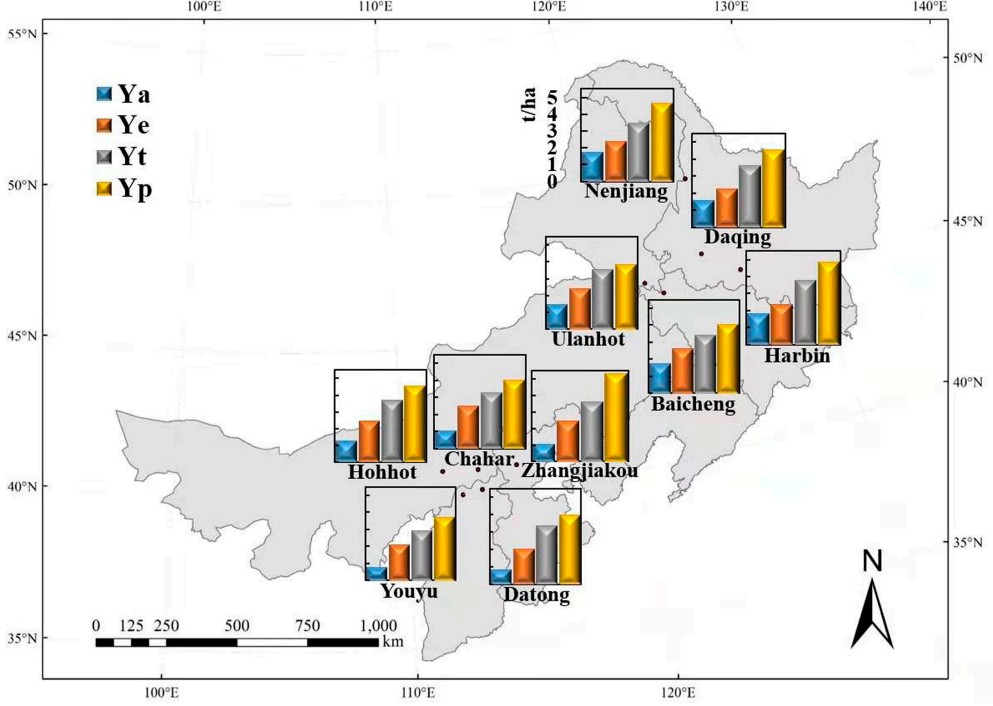

**Figure 4.** Spatial variations in potential yield (*Yp*), attainable yield (*Yt*), experimental yield (*Ye*), and actual yield (*Ya*) of rain-fed oat in North and Northeast China.

## 3.2. Attainable Yield, Experimental Yield, and Actual Yield

The mean *Yt* and *Ye* values of oat were 3369 kg/ha and 2317 kg/ha in NC and 3571 kg/ha and 2396 kg/ha in NEC, respectively (Table 2). *Yt* fluctuated from 2916 to 3691 kg/ha in NC and from 3416 to 3786 kg/ha in NEC. The *Ye* values varied from 2034 to 2490 kg/ha in NC and from 2265 to 2602 kg/ha in NEC. Adequate inputs, reasonable cultivation techniques, and high-quality oat varieties in experimental stations enabled *Yt* and *Ye* to be relatively high. In contrast, *Ya* was much lower, with its average value less than 2000 kg/ha (Table 2). The mean *Ya* values were only 989 kg/ha and 1656 kg/ha in NC and NEC, respectively, which ranged from 795 to 1230 kg/ha in NC and from 1462 to 1800 kg/ha in NEC.

Due to inappropriate fertilization, relatively old crop varieties, and low levels of agronomic management, farmers' *Ya* was significantly lower than *Yt* and *Ye*, which should be paid more attention if we want to close the yield gaps. In NC, higher *Yt*, *Ye*, and *Ya* values were mainly found in stations with higher latitudes (Zhangjiakou, Hohhot, and Chahar Cities, with values exceeding 3369 kg/ha, 2317 kg/ha, and 989 kg/ha, respectively) (Figure 4). However, in NEC, the spatial patterns of *Yt*, *Ye*, and *Ya* were relatively complicated. Higher *Yt* values were primarily found in Heilongjiang Province in NEC (Harbin and Daqing Cities, with values higher than 3571 kg/ha). Higher values of *Ye* were primarily found in Jinlin and Inner Mongolia of NEC (Baicheng and Ulanhot Cities, with values higher than 2396 kg/ha). Yet higher values of *Ya* were mainly distributed in Heilongjiang and Jilin Provinces of NEC, where have more abundant rainfall and relatively fertile soil (Figure 4).

*3.3. Different Levels of Yield Gaps*

In the study regions, the average values of YG-I were 935 kg/ha and 818 kg/ha, accounting for 21% and 18% of *Yp* in NC and NEC, respectively. YG-I varied from 643 kg/ha to 1697 kg/ha in NC, and from 285 kg/ha to 1182 kg/ha in NEC (Table 2). These differences were largely caused by the uncontrollable climatic conditions including solar radiation, sunshine duration, temperature, and precipitation. High values of YG-I were mainly distributed in the areas of Heilongjiang and Hebei Provinces. The greatest YG-I value was found in Zhangjiakou City of Hebei Province, which was significantly higher than that at the other stations. In NEC, high YG-I values were mainly found in Harbin, Nenjiang, and Daqing Cities, with each value exceeding 818 kg/ha (18% of *Yp*) (Figure 5).

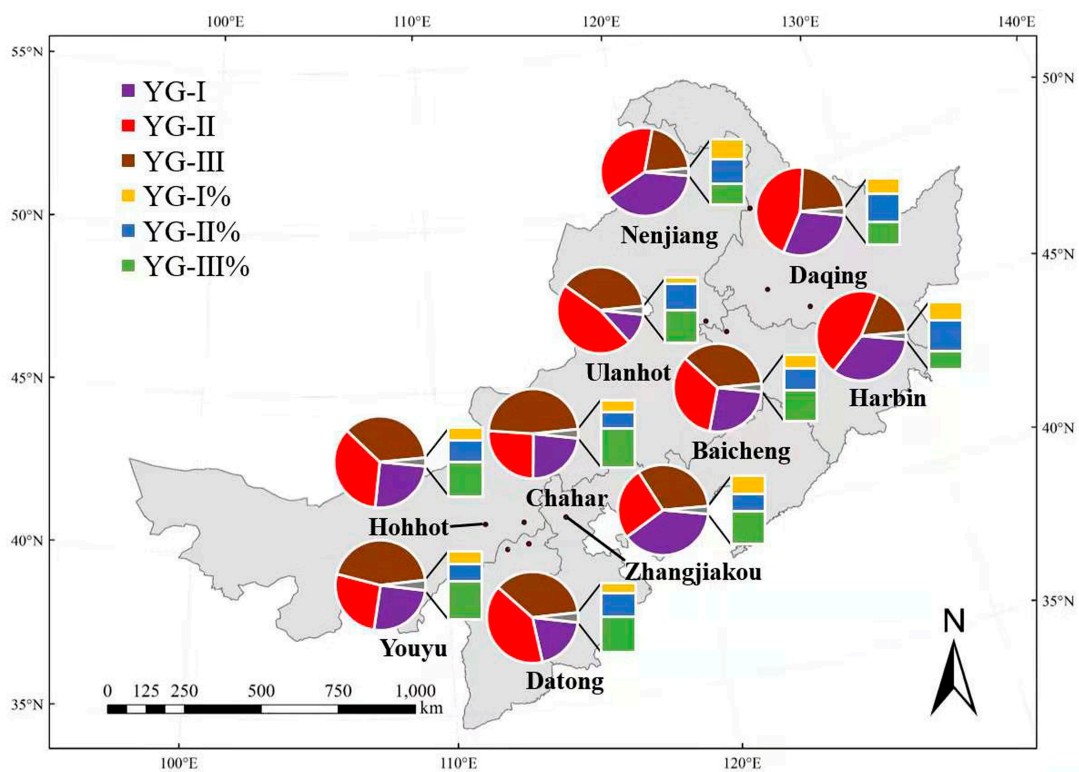

**Figure 5.** Yield gaps and yield gap percentages of rain-fed oat in North and Northeast China. The purple, red, and brown shadings indicate YG-I (yield gap I), YG-II (yield gap II), and YG-III (yield gap III), respectively. The yellow, blue, and green shadings indicate YG-I% (yield gap I percentage), YG-II% (yield gap II percentage), and YG-III % (yield gap III percentage), respectively.

The mean YG-II was 1051 kg/ha and 1175 kg/ha, and accounted for 31% and 33% of *Yt* in NC and NEC, respectively. YG-II varied between 786 kg/ha (YG-II percentage was 24%) and 1303 kg/ha (39%) in NC and between 814 kg/ha (24%) and 1435.55 kg/ha (38%) in NEC (Table 2). Higher values

of YG-II percentage were distributed in Zhangjiakou, Hohhot, and Datong Cities in NC, while in NEC, higher values were distributed in Harbin and Daqing Cities (Figure 5). Given the notable differences in soil type and texture among 10 experiment stations, the spatial variations in YG-II were mainly due to different soil nutrient and soil fertility. Although YG-II could be narrowed by optimizing farmers' fertilizer management and tillage practices, the improvement of soil quality was relatively hard to achieve in the short term.

The average values of YG-III were 1328 kg/ha in NC and 740 kg/ha in NEC, which accounted for 58% and 31% of *Ye*, respectively. YG-III varied from 1197 kg/ha to 1443 kg/ha in NC and from 551 kg/ha to 944 kg/ha in NEC (Table 2). The values of YG-III were higher in NC than in NEC in almost all sites. In NC, high values of YG-III (i.e., >1328 kg/ha) were distributed in Zhangjiakou and Chahar Cities, while in NEC, higher values (i.e., >740 kg/ha) were distributed in Baicheng and Ulanhot Cities (Figure 5). As YG-III was considered to be relatively easy to narrow yield gap in actual production compared with the other two yield gaps, and it could be reduced through improvement of agronomic practices, such as adopting more appropriate planting density, adjustment of sowing date, introduction of novel variety, and disease and insect control.

## 4. Discussion

### 4.1. Potential Yield under Rain-Fed Systems

The concept of potential yield is widely used in the field of yield gap research. In this study, the potential yield of oat is simulated using the AEZ model recommended by FAO. Compared with other models, although the AEZ model cannot simulate the effects of the crop growth process and cultivation management practices on crop yield, it is very suitable for evaluating the climatic potential productivity of crop. In addition, the AEZ model takes full account of multiple climatic factors affecting crop production, the required data are not difficult to be collected, and the corresponding parameters can be easily adjusted according to the characteristics of crop. Thus, despite the emergence of various crop models in recent years, the AEZ model remains an effective tool for calculating the crop potential productivity [22,23].

Potential yield is limited by temperature and solar radiation for irrigated systems, but under the rain-fed conditions, precipitation is also one of the most important limiting factors for potential yield. In this study, we defined the potential yield of oat in a rain-fed system, and the main driver of rain-fed oat yield is water [24]. Generally, the crop potential yield in rain-fed systems is much lower than that in irrigated systems, because water deficit is a major constraint for rain-fed crops during their growing season [11]. For instance, in India, the water-limited potential yield and water non-limiting potential yield of soybean were 2170 kg/ha and 3020 kg/ha respectively, indicating a 28% reduction in yield due to water deficit [25]. In our study, actual yields of rain-fed oat reach only 23% and 38% of potential yield in NC and NEC, respectively. Unlike rain-fed crops, irrigated rice yields are 78% of potential yield in China [26], and average wheat yields under irrigated systems can reach 80% of potential yield in northwest India [27]. It is necessary to propose effective methods to increase oat yields under non-irrigated systems. For instance, properly adjusting sowing date to improve the utilization of rainfall is an effective approach to increase yields and developing varieties with high photosynthetic efficiency and high water utilization through breeding methods can sufficiently adapt to climatic conditions of different regions, which helps to further explore yield potential.

### 4.2. Yield Gaps and Their Limiting Factors

This study quantified three levels of yield gaps and analyzed their spatial variations. Given the various limiting factors to yield gaps, the importance of each constraint to yield gaps may be different. YG-I is mainly restricted by uncontrollable climatic factors, and changes of temperature, solar radiation, and precipitation are not regular, which means these three limiting factors are equally important. However, YG-II is only affected by soil conditions. The differences in soil properties

lead to discrepancy in YG-II between the study sites. In our study, YG-III constraints mainly include management, variety, and profit. Management measures are easily optimized by farmers, which is considered to be the main factor restricting YG-III. The fluctuation of crop prices and production costs primarily affect the farmers' profit, determining the farmers' investment in production. Farmer's profit has an indirect impact on YG-III, so it is the least important restraint.

In addition, the difficulty of narrowing the three yield gaps is also differential. The potential yield is determined by the non-controllable climatic conditions (e.g., solar radiation and temperature) without regard to environmental costs or risks [28,29]. The attainable yield is obtained by optimizing various cultivation measures, which is not easy to be increased. The average YG-I (average YG-I percentage) in NC and NEC is less than the other two levels of yield gaps, reaching only 935 kg/ha (21%) and 818 kg/ha (18%), respectively. Therefore, it is the most difficult to narrow YG-I among the three yield gaps. YG-II is mainly determined by soil conditions, which can be reduced through the effective improvement in soil fertility, such as optimizing fertilizer management and tillage practices. However, improvement of soil quality is relatively hard to be realized in farmers' fields subject to high cost and slow effect.

In contrast, given that the limiting factors of YG-III are mainly the agronomic practices that are relatively easier to be optimized, it is feasible for farmers to increase their yields by narrowing the YG-III. There are many constraints on YG-III, among which the management measures of farmers play an indispensable role. In this study, we find that extensive and diversified management methods are the main reasons for the difference in yield between different farmers' fields and experimental stations, such as excessive rates of fertilizer application, unreasonable fertilization methods, high sowing rate, unsuitable sowing date, and mixed varieties. To understand farmers' actual yield limitations well, the fertilizer use, planting date, variety, and other factors should be taken into account [30]. The actual yield is usually much smaller than other kinds of yields, so it is advocated by the majority of researchers to grope for the restricting factors to actual yield in order to decrease yield gap in farmers' fields [31–33]. Particularly in the regions with low level of actual yields, the yield gaps have great potentials to be closed in the near future.

*4.3. Approaches to Narrow Yield Gaps*

An essential goal of studying yield gaps is to explore effective approaches to narrow yield gaps between farmers. It is significant to investigate feasible measures of narrowing yield gaps for ensuring regional food security. For the three levels of yield gaps, what we recommend is to narrow YG-III in actual production owing to higher YG-III percentage for most study sites. Since water deficit is a major constraint for the rain-fed oat, improving the utilization of rainfall is an effective approach to narrow the yield gaps. In the semi-arid area of Northern China, properly adjusting the crop sowing date in order to match it with the rhythm of precipitation during the growth season, is a widely adopted measure for improving water use efficiency [34,35]. Moreover, timely sowing is conducive to improve the utilization efficiency of local light and heat resources. For instance, it is suitable to sow early in spring oat-planting areas with warm climate, and the sowing date should be appropriately postponed in summer oat planting area where the climate is cold, which contributes to increasing seed setting rate [36]. Therefore, it is an effective agronomic practice to narrow the yield gaps of oat by appropriately adjusting sowing date according to regional rainfall pattern and photo-thermal resource.

In addition, integrated fertilizer management is also an effective way to increase the yield of oat [37,38]. Nitrogen is considered to be the main fertilizer in oat cultivation [39,40]. Oat yields increase with increasing amounts of nitrogen fertilization within a certain range [41–43], but after reaching a certain high nitrogen rate, increasing nitrogen application rate did not have much influence on oat yield [44]. Consequently, the yield gaps between different regions can be narrowed if farmers are guided to apply fertilizer according to soil nutrient and soil fertility in different ecological regions. Planting density is also a limiting factor to farmers' actual yield [45]. The traditional high sowing rate (200–300 kg/ha) not only wastes the amount of seeds, but also fails to achieve high yield. The seeding

rate used in study sites is 150 kg/ha, which is much lower than in farmers' fields, but the yield is higher than farmers' actual yield. Farmers should adjust local agronomic practices to reduce seeding rate properly. Besides, the yield-increasing effect of the new crop varieties should not be neglected [46]. Genetic selection with optimal crop management can increase yields of major cereals under rain-fed conditions [47]. We should fully exploit the diversity of oat varieties and select suitable varieties based on local climate and soil suitability. The farm profit can also affect the farmers' investments in production [48]. The fluctuation of oat prices and the increase of production costs directly have an effect on the income of farmers, which may undermine the enthusiasm of farmers for production. By improving technology and optimizing management, the production costs can be reduced to narrow yield gaps and increase farmers' income.

## 5. Conclusions

This study put forward a conceptual framework to analyze the different dimensions of yield gaps. We quantified four kinds of yields (*Yp*, *Yt*, *Ye*, and *Ya*) and three levels of yield gaps, and revealed their spatial variations for rain-fed oat in NC and NEC. The spatial distribution of potential yield decreased from south to north and from east to west, which is mainly determined by climatic factors including light, temperature, and precipitation. Given the great differences in the spatial patterns and limiting factors for the three levels of yield gaps, we should take differentiated measures to reduce the yield gaps in different regions.

The climatic and agronomic dimensions of yield gaps were diagnosed and various possible approaches to narrow yield gaps were discussed. Among the three levels of yield gaps, it is most difficult to narrow YG-I due to the limitation of non-controllable climatic dimensions. Despite that YG-II could be reduced through improvement of soil fertility, it is still not so easy to improve soil quality in the short term. Compared to YG-I and YG-II, narrowing YG-III is the most practical for farmers. It can be reduced by improving the agronomic practices, such as adjusting sowing dates, optimizing planting density, and selecting high-yielding varieties. In summary, research on yield gaps caused by climatic conditions and agronomic practices will have a positive effect on increasing oat yields and farmers' income in the near future.

**Author Contributions:** Q.C. and Y.L. obtained funding and revised the manuscripts. Z.Z. verified data analysis. C.W., X.S., and T.Z. designed the study and collected related data. C.W., J.L., and S.L. analyzed the data and described the results. C.W. wrote the original draft.

**Funding:** This work was supported by the National Natural Science Foundation of China (31871581; 31801315); the National Key Research and Development Program of China (2016YFD0300201); and the National Oat Industry Technology System in China (CARS-08).

**Acknowledgments:** The authors would like to thank the National Oat Industry Technology System in China for providing data and the related information. We thank local agricultural investigators who helped with collecting yield data. Our special thanks are due to China Agricultural University for support to the study.

**Conflicts of Interest:** The authors declare no conflict of interest.

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
