# Peer review of "Diagnosing the Climatic and Agronomic Dimensions of Rain-Fed Oat Yield Gaps and Their Restrictions in North and Northeast China"

_sustainability, doi:10.3390/su11072104_

Round 1

Reviewer 1 Report

Evaluation and comments to manuscript No. sustainability-467157 entitled “Diagnosing the climatic and agronomic dimensions of rain-fed oat yield gaps and their restrictions in North and Northeast China”.

Authors: Chong Wang et al. 

Currently, studies of this type are very important, and I confirm that “confronted with the great challenges of globally growing populations and food shortages, society must achieve future food security by increasing grain output and narrowing the gap between potential yields and farmers’ actual yields”. The manuscript is primarily new data and a new look at the above issue. Therefore, in my opinion, these results may be of interest to the readers of the Sustainability  journals.

Overall, the experimental design and data analysis are appropriate, and the introduction is consistent enough, with the exception of the discussion, which is poor. Therefore, the work should make a good contribution to the literature.

I did not notice any major factual or editorial errors at work. On the other hand, I do not mention any minor errors, because they are in some sense something normal in this type of work and they do not affect the general perception of it.

My suggestions to authors

Study area –  please add coordinates to the test sites.

Figs. 1, 3, 4 and 5 are hard to read, for example, they should be enlarged before printing.

In my opinion, the discussion is quite poor as a regular article. Therefore, please enrich this section.

My conclusion

In my opinion, the work should be accept and published after minor revision. However, all changes are vital before publication of the MS, especially the discussion must be improved. Otherwise, the MS can not be published.

Author Response

Response to Reviewer 1 Comments

Point 1:

Comments and Suggestions for Authors

Currently, studies of this type are very important, and I confirm that “confronted with the great challenges of globally growing populations and food shortages, society must achieve future food security by increasing grain output and narrowing the gap between potential yields and farmers’ actual yields”. The manuscript is primarily new data and a new look at the above issue. Therefore, in my opinion, these results may be of interest to the readers of the Sustainability journals.

Overall, the experimental design and data analysis are appropriate, and the introduction is consistent enough, with the exception of the discussion, which is poor. Therefore, the work should make a good contribution to the literature.

I did not notice any major factual or editorial errors at work. On the other hand, I do not mention any minor errors, because they are in some sense something normal in this type of work and they do not affect the general perception of it.

Response 1:

Thank you very much for your valuable comments on our manuscript. Those comments and suggestions are very helpful for improving the quality of our paper and researches. We have studied all the comments carefully and made corrections using the "Track Changes" function in the uploaded revised version.

To enrich our paper and highlight the novelty of study, as you suggested, we have added some sentences in the Discussion section. (Please see the changed content in the Discussion section).

The main corrections and the responses to your comments are as follows:

Specific Comments Points & Our Responses:

1) Study area – please add coordinates to the test sites.

>> We added coordinates to ten study sites. (Line 89-92 in the revised version)

2) Figs. 1, 3, 4 and 5 are hard to read, for example, they should be enlarged before printing.

>> We enlarged Figs. 1 and 3, and remapped Figs. 4 and 5. (Line 105, 230, 265, 290 in the revised version)

3) In my opinion, the discussion is quite poor as a regular article. Therefore, please enrich this section.

>> Thank you for your suggestion. we have made carefully modification in the Discussion section. (Line 309, 327-335, 338-349, 355-356, 363-372, 377-378, 383-386, 388-389, 391-394, 396-408 in the revised version)

My conclusion

In my opinion, the work should be accepted and published after minor revision. However, all changes are vital before publication of the MS, especially the discussion must be improved. Otherwise, the MS can not be published.

>> Thanks a lot for your kind suggestion. In order to improve the quality of paper, we have enriched the Discussion section (Please see the changed content in the Discussion section).

Review Report Form (Reviewer 1)

Open   Review

()   I would not like to sign my review report

(x)   I would like to sign my review report

English   language and style

(   ) Extensive editing of English language and style required

(   ) Moderate English changes required

(x)   English language and style are fine/minor spell check required

(   ) I don't feel qualified to judge about the English language and style

Yes

Can   be improved

Must   be improved

Not   applicable

Does   the introduction provide sufficient background and include all relevant   references?

(x)

( )

( )

( )

Is   the research design appropriate?

(x)

( )

( )

( )

Are   the methods adequately described?

(x)

( )

( )

( )

Are   the results clearly presented?

( )

(x)

( )

( )

Are   the conclusions supported by the results?

(x)

( )

( )

( )

Comments and

Suggestions for   Authors

Currently,   studies of this type are very important, and I confirm that “confronted with   the great challenges of globally growing populations and food shortages,   society must achieve future food security by increasing grain output and   narrowing the gap between potential yields and farmers’ actual yields”. The   manuscript is primarily new data and a new look at the above issue.   Therefore, in my opinion, these results may be of interest to the readers of   the Sustainability journals.

Overall,   the experimental design and data analysis are appropriate, and the   introduction is consistent enough, with the exception of the discussion,   which is poor. Therefore, the work should make a good contribution to the   literature.

I   did not notice any major factual or editorial errors at work. On the other   hand, I do not mention any minor errors, because they are in some sense   something normal in this type of work and they do not affect the general   perception of it.

Submission   Date

04   March 2019

Date   of this review

05   Mar 2019 14:08:54

Reviewer 2 Report

The manuscript was well written except some few oversight by authours. I made some comments to help improve the quality of the manuscript. Please find attached the manuscript with my  comments.

Author Response

Response to Reviewer 2 Comments

Point 1:

Comments and Suggestions for Authors

The manuscript was well written except some few oversight by authors. I made some comments to help improve the quality of the manuscript. Please find attached the manuscript with my comments.

Response 1:

Thank you very much for your insightful comments concerning our manuscript. Those comments are very valuable and helpful for improving the quality of our paper, as well as the significance of this research. We have studied all the comments carefully and have made corresponding corrections using the "Track Changes" function in the revised version.

The main corrections and the responses to your comments are as follows:

Point 2:

Some minor comments on English language errors shown in the manuscript:

Line71-72: ‘we attempt to explore the oat yield gaps’ replaced by ‘we examined oat yield gaps’. (Line 76 in the revised version)

Line73: ‘are’ replaced by ‘were’. (Line 77 in the revised version)

Line155: ‘include’ replaced by ‘included’. (Line 183 in the revised version)

Line158: ‘include’ replaced by ‘included’. (Line 186 in the revised version)

Responses to the other comments concerning the logical and scientific issues:

1) What about the genetic or inherent component of potential yield? This is very important. You only highlighted the environmental component. Find a way of adding the genetic component in the sentence.

>> Thanks a lot for your kind suggestion. As suggested, environmental and genetic components together affect potential yield, so we have added the genetic component of potential yield in the sentence. (Line 58-61 in the revised version)

2) Do NC and NEC have different climatic conditions?

>> There are different climatic conditions during oat growing season at these ten sites in NC and NEC. We have added different climatic conditions in the sentence, including ranges of average temperature, maximum temperature, minimum temperature, precipitation, solar radiation, and sunshine duration. (Line 93-103 in the revised version)

3) Your formula for YG-I percentage is inaccurate. You were supposed to multiply the fraction of the yield difference by 100. That is, [(Yp-Yt)/Yp]*100

>> Based on your suggestions, we have rewritten these three formulas for YG-I percentage, YG-II percentage, and YG-Ⅲ percentage, respectively. (Line 163, 165, 167 in the revised version)

4) YG-I and YG-II above have no equation number?

>> We have added equation numbers of YG-I and YG-II, revised equation number of YG-Ⅲ according to your advice. (Line 163, 165, 167 in the revised version)

Review Report Form (Reviewer 2)

Open   Review

(x)   I would not like to sign my review report

(   ) I would like to sign my review report

English   language and style

(   ) Extensive editing of English language and style required

(   ) Moderate English changes required

(x)   English language and style are fine/minor spell check required

(   ) I don't feel qualified to judge about the English language and style

Yes

Can   be improved

Must   be improved

Not   applicable

Does   the introduction provide sufficient background and include all relevant   references?

(x)

( )

( )

( )

Is   the research design appropriate?

( )

( )

( )

(x)

Are   the methods adequately described?

( )

(x)

( )

( )

Are   the results clearly presented?

(x)

( )

( )

( )

Are   the conclusions supported by the results?

(x)

( )

( )

( )

Comments and

Suggestions for   Authors

The   manuscript was well written except some few oversight by authors. I made some   comments to help improve the quality of the manuscript. Please find attached   the manuscript with my comments.

Submission   Date

04   March 2019

Date   of this review

10   Mar 2019 16:17:16

Reviewer 3 Report

The subject of the manuscript is interesting and present the topic which is very important for agriculture and food production at regional level.

Some issues should be described in more detailed way, especially more details should be provided for evaluation of each type od the yield.

1) What was the source of climate data for evaluation of Yp? It was based on real meteorological measurements (and then interpolated) or it is based on meteorological models. From which period these data were included?

2) It would be good if some important details about experiments form which the Ye was estimated. I think that average rates of NPK fertilization,  basic information about crop protection, typical sowing rate and sowing date.

3) The authors “adopted the highest experimental yield as attainable yield”. Some details should be provided. Is it the highest yield on the basis of the results from individual plot? In some circumstances it can be very high.

4) Some basic information about crop management used by farmers should be added (the same as mentioned in point 2 above).

5) There is lack information about effect of soil conditions on the yield gaps. Is there soil quality the same in experiments as in farms? Oat is very often in farms cultivated on poorer soil in comparison with other species like e.g. wheat. Was the effect of various soil conditions included in the evaluation of Ye and Ya?

6) Table 1 presents min and max. What is min and max in case of each type of yield? At what geographical level min and max were estimated? Depending of geographical level min and max will be different.

It wold be more interesting if the authors present the rank of factors which are most important in yield gaps, especially in YG-III.

Author Response

Response to Reviewer 3 Comments

Point 1:

Comments and Suggestions for Authors

The subject of the manuscript is interesting and present the topic which is very important for agriculture and food production at regional level.

Some issues should be described in more detailed way, especially more details should be provided for evaluation of each type of the yield.

Response 1:

Thank you very much for your valuable comments on our manuscript. Those comments and suggestions are very helpful for improving the quality of our paper and researches. We have studied all the comments carefully and made corrections using the "Track Changes" function in the uploaded revised version.

In the Materials and Methods section, we have added some details about experiments, basic information on farmers’ crop management, (Line 191-199, 201-209 in the revised version) and soil conditions of study sites (Table 1 in the revised version). For evaluation of each type of the yield, we have added details to some sentences that were not clearly expressed in the original version. (Line 137-139, 146-148, 152-155, 158-160 in the revised version)

The main corrections and the responses to your comments are as follows:

Specific Comments Points & Our Responses:

1) What was the source of climate data for evaluation of Yp? It was based on real meteorological measurements (and then interpolated) or it is based on meteorological models. From which period these data were included?

>> The climate data are based on real meteorological measurements, provided by the National Meteorological Information Centre of China Meteorological Administration (CMA), which are mentioned in the original manuscript (Line 181-183 in the revised version). The time period of the climate data is between 2011 and 2015, which is added in the context. (Line 182-183 in the revised version)

2) It would be good if some important details about experiments form which the Ye was estimated. I think that average rates of NPK fertilization, basic information about crop protection, typical sowing rate and sowing date.

>> As suggested, we have added some important details about experiments, (Line 191-199 in the revised version) in which the information about the sowing date and harvest date is presented in a table. (Table 1 in the revised version)

3) The authors “adopted the highest experimental yield as attainable yield”. Some details should be provided. Is it the highest yield on the basis of the results from individual plot? In some circumstances it can be very high.

>> Sorry for the confusing expressions. The maximum experimental yield in ten study sites from 2011 to 2015 is regarded as attainable yield. The highest experimental yield is based on the results from experiment, not individual plot, which represents the maximum yield level of the experiment. We added a sentence as a supplement in the Materials and Methods section. (Line 146-148 in the revised version)

4) Some basic information about crop management used by farmers should be added (the same as mentioned in point 2 above).

>> Thanks for your kind suggestion. We have added basic information about farmers’ crop management in the Materials and Methods section. (Line 201-209 in the revised version)

5) There is lack information about effect of soil conditions on the yield gaps. Is there soil quality the same in experiments as in farms? Oat is very often in farms cultivated on poorer soil in comparison with other species like e.g. wheat. Was the effect of various soil conditions included in the evaluation of Ye and Ya?

>> We have added basic information of soil conditions of the ten study sites in the Materials and Methods section, which is presented in a table (Table 1 in the revised version). Ye and Ya are obtained at the same site, and soil conditions are the same for each location. Therefore, effects of soil conditions are not included in the evaluation of Ye and Ya.

6) Table 1 presents min and max. What is min and max in case of each type of yield? At what geographical level min and max were estimated? Depending of geographical level min and max will be different.

>> The maximum and minimum yield for each type of yield was noted in the table 2. (Table 2 in the revised version). We add some sentences to explain the question that maximum and minimum values for each type of yield are estimated. (Line 249-252 in the revised version)

It would be more interesting if the authors present the rank of factors which are most important in yield gaps, especially in YG-III.

>> Thanks a lot for your kind suggestion. We have added the rank of factors which are most important in yield gaps to improve the quality of the paper. (Line 338-347 in the revised version)

Review Report Form (Reviewer 3)

Open   Review

(x)   I would not like to sign my review report

(   ) I would like to sign my review report

English   language and style

(   ) Extensive editing of English language and style required

(   ) Moderate English changes required

(   ) English language and style are fine/minor spell check required

(x)   I don't feel qualified to judge about the English language and style

Yes

Can   be improved

Must   be improved

Not   applicable

Does   the introduction provide sufficient background and include all relevant   references?

(x)

( )

( )

( )

Is   the research design appropriate?

( )

( )

( )

(x)

Are   the methods adequately described?

( )

( )

(x)

( )

Are   the results clearly presented?

( )

( )

(x)

( )

Are   the conclusions supported by the results?

( )

( )

(x)

( )

Comments and

Suggestions for   Authors

The   subject of the manuscript is interesting and present the topic which is very   important for agriculture and food production at regional level.

Some   issues should be described in more detailed way, especially more details   should be provided for evaluation of each type od the yield.

Submission   Date

04   March 2019

Date   of this review

13   Mar 2019 16:23:22

Round 2

Reviewer 3 Report

The authors included all the comments in the edited manuscript. In my opnion changes which have been done by the authors are sufficient and manuscript can be published in present form.